# STIMGRASP: A Home-Based Functional Electrical Stimulator for Grasp Restoration in Daily Activities

**DOI:** 10.3390/s23010010

**Published:** 2022-12-20

**Authors:** Renato G. Barelli, Valter F. Avelino, Maria Claudia F. Castro

**Affiliations:** Electrical Engineering Department, Centro Universitário FEI, São Bernardo do Campo 09850-901, SP, Brazil

**Keywords:** functional electrical stimulation (FES), circuit design, circuit application, grasping, tetraplegia, hemiplegia

## Abstract

Thousands of people currently suffer from motor limitations caused by SCI and strokes, which impose personal and social challenges. These individuals may have a satisfactory recovery by applying functional electrical stimulation that enables the artificial restoration of grasping after a muscular conditioning period. This paper presents the STIMGRASP, a home-based functional electrical stimulator to be used as an assistive technology for users with tetraplegia or hemiplegia. The STIMGRASP is a microcontrolled stimulator with eight multiplexed and independent symmetric biphasic constant current output channels with USB and Bluetooth communication. The system generates pulses with frequency, width, and maximum amplitude set at 20 Hz, 300 µs/phase, and 40 mA (load of 1 kΩ), respectively. It is powered by a rechargeable lithium-ion battery of 3100 mAh, allowing more than 10 h of continuous use. The development of this system focused on portability, usability, and wearability, resulting in portable hardware with user-friendly mobile app control and an orthosis with electrodes, allowing the user to carry out muscle activation sequences for four grasp modes to use for achieving daily activities.

## 1. Introduction

According to the World Health Organization (WHO), the estimated global annual incidence of spinal cord injury (SCI) is 40 to 80 cases per million, while stroke is one of the leading causes of death and long-term disability [1,2,3]. Both are life-disrupting situations due to motor limitations and other health conditions, imposing restrictions on productive, personal, and social subjects’ lives. Motor deficits can affect the movements of the upper and lower limbs. However, since autonomy and independence restoration are the main objectives of rehabilitation programs, an effort has been made to restore hand movements to support the execution of activities of daily living (ADLs), such as feeding, dressing, writing, and others involving grasping. Even so, especially in low- and middle-income countries, the availability of quality assistive devices and rehabilitation services is minimal, and opportunities for participation in personal and social life are limited.

Functional electrical stimulation (FES) is the application of electrical stimuli to assist the functional movements used in daily activities. Doucet et al. [4] and Marquez-Chin and Popovic [5] presented reviews of the fundamentals of FES technology, its applications, and a brief description of the potential future, Ragnarsson [6], Rupp et al. [7] and Venugopalan et al. [8] gave an overview of FES devices, while Kapadia et al. [9] described the protocols and clinical results used to retrain upper-extremity function for over 15 years, assisting over 200 patients with either stroke or SCI. The results showed more significant improvements in the self-care subscores of the Functional Independence Measure compared to individuals who received conventional occupational therapy. An FES system used as assistive technology is referred to as a neuroprosthesis. Its main components for grasping restoration include an electric stimulator, a user–machine interface, electrodes, and an orthosis that provides additional mechanical assistance and helps to fix the electrodes. The most common restored patterns are lateral grasp (also called key grasp) and palmar grasp (also called cylinder or power grasps) [4,5,6,7].

The Freehand invasive system from the 1980s [6,10,11] introduced one of the most successful rehabilitation programs for grasp restoration. It was commercialized from 1994 to 2001 by NeuroControl Corp., the United States, and applied to 200 people [8]. Despite being more functional than a noninvasive system, it faced some difficulties related to the cost and safety regarding the surgical procedure and some breakdown issues with the shoulder position sensor and the external transmitter coil [8,11].

The literature also showed several examples of noninvasive systems from the 1990s and 2000s. The Bionic Glove, created for people with hemiplegia and C6–C7 tetraplegia, used a position sensor to detect voluntary wrist flexion/extension movements to trigger the stimulation sequence for hand opening and closing. Conductive areas on the internal surface of the fingerless neoprene glove made contact with self-adhesive electrodes previously placed on the skin [12]. The Bionic Glove was commercialized by Neuromotion Inc., Canada, as the Tetron Glove until 1999, and then the IP was acquired by Rehabtronics Inc., Canada [8], and is now commercialized as ReGrasp. This device is similar to the previous, except it uses a Bluetooth earpiece to control the FES device using head movements.

The NESS Handmaster was a hybrid system combining a wrist/hand ergonomic orthosis housing the electrodes and providing wrist stabilization with a three-channel FES system attached with a cable. The system was designed especially for people with C5 tetraplegia, those with a loss of voluntary wrist extension activity, and stroke survivors for exercise, using electrodes fixed to the user’s orthosis [13]. This system was improved with RF communication and is now commercialized as the H200 Wireless System by Bioness Inc., the United States.

The Compex Motion stimulator was a four-channel multipurpose programmable FES system with arbitrary stimulation sequences controlled in real-time by an external sensor. A graphical user interface software installed on a personal computer (PC) assisted the system configuration saved on a chip card inserted into the stimulator [14,15].

The Motionstim 8 (Krauth + Timmermann, Germany) was an eight-channel portable FES system for therapeutic interventions, used by Rupp et al. [16] as assistive technology. They introduced an easy-to-handle forearm electrode sleeve, similar to the Bionic Glove, addressing one of the biggest complaints and time-consuming tasks faced during the daily use of FES, which is the difficulty in electrode positioning. Therefore, quick and accurate surface electrode positioning is a challenging prerequisite for a noninvasive FES system application as assistive technology [17]. To date, the manufacturer commercializes only up to four-channel EMG-triggered devices. An eight-channel FES device, the MyndMove, is provided by MyndTech Inc., Canada. This device has embedded simulation protocols that can help the user with over 30 reach and grasp functions and has an intuitive user interface that allows the therapist to select and deliver a personalized program therapy.

### Basic Stimulator Architecture

Figure 1 shows the main block diagram of a noninvasive stimulator unit in which the stimulus is applied by surface electrodes. Below is a brief review of this architecture.

The type of application expected for the stimulator defines the control interface. Systems that target the final user commonly apply interfaces in the equipment, such as displays, potentiometers, and buttons, or alternatively may use position sensors and myoelectric signals (EMG) to initiate the movements whose parameters are already preconfigured, as mentioned before [8,14,16,18,19,20]. Recent advances have also shown the use of brain–machine interfaces (BCI) as user intention in FES systems [7,21,22,23,24,25]. However, the two neuroprostheses available on the market that are currently known use wireless communication (Bioness—radio frequency and ReGrasp—Bluetooth), buttons on the control unit (both), and head movements from one earpiece (ReGrasp).

They are also battery-powered systems, providing mobility and ensuring a lower risk of electric shock for the patient. As batteries present low voltages and a voltage reduction over time, this power method requires a buck–boost converter to keep the voltage stable and in the appropriate interval to power the different parts of the circuit. In addition, voltage levels to generate the necessary current amplitudes to stimulate the human tissue are usually superior to those applied to power the electronic circuits, requiring an increase of the voltage levels that power the output circuit to generate stimulation pulses.

A voltage increase can be achieved through some methods, including boost transformers at the output stage—a topology that is still often used, despite increasing the size and weight of the circuit as well as causing electromagnetic interference [26]. A more modern topology is the use of switch boost converters. The system proposed by Qu et al. [27] presented a boost converter developed to generate 100 V from 12 V of input, operating at 50 kHz, while Qu et al. [28] showed a converter with the microcontroller as a control unit, reading the output voltage through a voltage divider as feedback. It was also possible to obtain more features with the microcontroller, such as a ramp initialization, with the progressive expansion of the pulse width. The stimulator, proposed by Xu et al. [29], used a DC/DC converter based on a commercial gated oscillator boost controller module (MCP1651 from Microchip^®^) with an external transistor and energy feedback, generating 30 V from 5 V of input. Similarly, the system proposed by Malešević et al. [18] used the step-up MAX773 integrated circuit (Maxim^®^), generating 94 V for the output stage.

The pulse generator circuit produces the output pattern waveform at a low power. It usually corresponds to an oscillating circuit or microcontroller whose output pulses have controllable characteristics regarding waveform, amplitude, pulse width, and frequency, defined through the interface with the user [18,27,28,30,31,32]. Symmetric biphasic constant current pulses have been preferred, preventing charge accumulation harmful to the subject and with better dynamic responses to the electrode–skin impedance variations, resulting in more predictive and repetitive muscular responses. However, a downside of constant-current stimulation is the possibility of causing skin burns due to a current density increase above 0.5 mA/cm${}^{2}$ in residual contact points when the electrode is released. However, according to Patriciu et al. [33], skin burns depend on the stimulus charge, which is more critical in applications using pulse widths on the order of ms and frequencies above 100 Hz. For the upper limbs, FES pulse widths are less than 600 µs and the frequencies are less than 50 Hz, resulting in small charges, and a good electrode–skin contact minimizes risks.

Current amplitudes from 20 to 40 mA are usually enough for upper limbs. The most commonly used frequencies range from 20 to 40 Hz to minimize muscle fatigue, increase comfort, and ensure a strong enough contraction resulting in joint movement. Furthermore, pulse widths from 250 to 600 µs are also used. However, some researchers have reported a superior selectivity and differences in the recruitment characteristics with narrow pulses, while wide pulses have more penetration power with stronger contractions. [4,5,9,16].

The modulation module usually modifies these pulses to generate a satisfactory waveform to stimulate the tissue after its amplification. Examples are amplitude or pulse-width modulations with a trapezoidal package so that the load delivered to the muscle increases or decreases gradually without steep variations. A stimulus gradation provides a greater comfort to the user and results in smoother movement transitions closer to physiological ones. In addition, it is suitable for applications where the user has an increased muscle tone, such as in hypertonia or spastic musculature [4,16,28,34].

Finally, the output stimulation channels are the circuits producing waveforms to apply to the skin through electrodes, using the high voltage generated by the boost converter and pulses generated at low power. Souza et al. [26] presented a systematic review of output stage circuits, revealing topologies such as step-up transformers, current mirrors, and H-bridge switching circuits, voltage-to-current converters, and others. Khosravani et al. [30] used a single Wilson current mirror to clamp the output current of all channels, while unique H-bridge circuits provided biphasic pulses for each channel. In the project proposed by Qu et al. [27], an H-bridge topology with four controlled switches and two voltage-controlled current sources, each one implemented with an operational amplifier and a MOSFET transistor, was applied for each channel, providing regulated output biphasic current. Instead, in [28], the H-bridge had two switches and two voltage-controlled current sources. Malešević et al. [18] also implemented an H-bridge output circuit; however, no detail was given. The circuit proposed by Masdar et al. [31] had a voltage-to-current converter based on a power operational amplifier (op-amp) in the output stage. Instead, in [32], the voltage-to-current converter included two coupled transconductance amplifiers based on a Holland structure, those determining the current of two complementary Wilson current mirrors followed by an H-bridge circuit. In this case, the transconductance amplifier contained a low-power op-amp.

Under this perspective, this study presents the development of the STIMGRASP, a new functional electrical stimulation system focusing on hardware development. The system consists of a neuroprosthesis for people with tetraplegia and hemiplegia. In addition to the stimulator, an orthosis with electrodes, PC software for the health professional, and a mobile app for the user to control the neuroprosthesis are included in the system, providing four sequences of muscular activation: hand opening, palmar and lateral grasping, and forefinger extension. The unique characteristics of the developed system are its portability, functionality, and wearability, enabling its application as assistive technology in ADLs.

## 2. Materials and Methods

The main component of this project presented in Figure 2 is the stimulator associated with an orthosis housing the electrodes, configured and controlled by a Windows platform and a mobile app.

### 2.1. Specifications

Considering the STIMGRASP daily use application as an assistive technology, the project specifications favored portability, wearability, and usability criteria, with the following main electrical characteristics.

1.PortabilityMiniaturization: the proposed stimulator must have its dimensions and weight reduced to be moved by the user without effort, which is important for functionality and discretion as an assistive device.Battery life: the circuit consumption and battery capacity should allow one to use the system during the day and charge it at night.2.WearabilityEasiness to wear the equipment: the more tasks the individual can carry out without aid from other people, the better their self-esteem will be. Thus, there is a need to guarantee user independence to don and doff the system.3.UsabilityUser-friendliness: the control interface must be user-friendly, considering the user’s motor limitations, especially for people with tetraplegia due to the lack of hand dexterity or even the impossibility of controlling finger movements.4.CharacteristicsMicrocontrolled 16-bit architecture, PIC24 series system.USB or Bluetooth communication (selectable).Symmetric biphasic constant current waveform.Independent stimulation outputs up to 8 channels.Stimulation parameters in commonly used range.Powered by rechargeable lithium-ion battery.LEDs’ general status indication.

### 2.2. Circuit Design

The circuit design described in detail in the following sections meets the block diagram shown in Figure 3.

#### 2.2.1. Power Sources

STIMGRASP uses several power sources. An NCR18650A rechargeable lithium-ion battery from Panasonic^®^, whose nominal voltage is 3.7 V (+VIN) and charge capacity is 3100 mAH with a protection circuit, is the main power of the STIMGRASP. It has an internal battery charger implemented from the MCP73837 model (Microchip^®^) with a USB connection.

The +3.3 V power unit is based on the buck–boost TPS63031 regulator (Texas Instruments^®^), suitable for lithium-ion battery applications. This power unit supplies the power to logic circuits, which consist mainly of the microcontroller, Bluetooth communication module, digital–analog converter (DAC), EEPROM, and status LEDs.

The +12 V power unit uses the MAX1523 boost converter (Maxim Integrated^®^). This power unit’s main charges are two switch units of +/−48 V, which demand the supply of a high voltage pulse circuit, and the −12 V unit. The switch element of the +12 V power unit is the N FDT86113 MOSFET-type transistor (Fairchild^®^), selected for this application due to its low resistance between drain and source and adequate maximum VDS voltage (100 V). The −12 V power unit uses a charge-pump voltage converter based on the TC7662 controller (Microchip^®^) and supplies, together with +12 V, a symmetrical power to an operational amplifier, whose voltage consumption is low and does not need excellent regulation. The +48 V power unit has the same circuit as the one used for the +12 V power unit, while the −48 V power unit uses the MAX776 controller (Maxim Integrated^®^) in an adjustable voltage configuration at the output. Aiming at saving energy, the +12 V, +48 V, and −48 V power units have enable signals (SHDN) controlled by the microcontroller.

#### 2.2.2. Biphasic Pulse Generation

A Howland current source circuit enables the generation of a biphasic current with adequate stabilization in the time interval (3 ms) necessary for switching each output stimulation channel. The circuit used to increase the output current capacity, presented in Figure 4, consists of a high voltage, precision, operational amplifier (ADA4700-1 from Analog Devices^®^) and two power Darlington transistors (MJD122 and MJD127). In this circuit, the Iout current is proportional to the Vi input voltage.

As the human skin tissue is not purely resistive but has resistive and capacitive characteristics [35,36], the output of the current source circuit may present delays when increasing or decreasing output pulses. Thus, a derivative compensation negative feedback (C9/R11 and C53/R49) accelerates the circuit response, correcting the system response to a critically damped behavior.

The microcontroller controls the DAC converter (MCP4725 from Microchip^®^) through I2C communication (400 kHz), which generates a biphasic waveform with an average threshold of +1.65 V Figure 5. This signal has eight multiplexed output channels, which are synchronized by the microcontroller with the switching of solid-state relays, allowing the sequencing of the stimulation channels. The signal generated by the DAC converter goes through a voltage follower and a voltage subtractor, whose reference is half of the +3.3 V voltage of the circuit. Thus, the voltage signal injected into the current source circuit (Vi) has a biphasic waveform with a maximum amplitude of +/−1.65 V.

#### 2.2.3. Output Channel Multiplexing

A biphasic pulse (20 Hz) stimulation was adopted. As the total pulse width is 600 µs (300 µs for each phase), the time interval between each pulse of the output channel is 49.4 ms. The strategy for generating eight output switching channels from one current source consists of distributing the pulses of the other seven channels in this time interval. Figure 6 shows the circuit that distributes the output signal from the pulse generation circuit, using a 74HC138 demultiplexer that receives the channel code from the microcontroller and activates each CPC2030N (IXYS^®^) solid-state relay.

Each solid-state relay has two identical and independent circuits in the same package. Its control circuit consists of optical coupling and bipolar output formed by 2 MOSFET transistors, enabling a current control up to 120 mA. Its dielectric strength is 1500 Vrms, while the output transistors operate with voltages up to 350 V with a series resistance of 30 Ω. The maximum relay turn-on/turn-off times are 1.5/0.4 ms. Adding these times to the pulse width can verify that the minimum time window for the output pulse of one channel is 2.5 ms. Therefore, mathematically, within the 50 ms time window (corresponding to a 20 Hz frequency), it would be possible to allocate up to 20 independent channels in a multiplexing sequence instead of the eight channels of the STIMGRASP. Due to the switching between channels, there may be instants when no load is present in the current source output (interval in which the solid-state relays are commuting). In this condition, the current source output may be unstable, and to avoid this situation, a circuit connects a fixed load of 1 kΩ in the circuit output of ADA4700-1.

#### 2.2.4. Data Communication Channels

STIMGRASP uses a Bluetooth serial port module (HC-06) directly connected to the UART2 port of the microcontroller. Once this module and the master device are paired, whether a smartphone or computer, the communication is transparent, and no signal treatment is necessary. A USB–serial CP2102 (Silicon Lab.^®^) converter is also part of the circuit, enabling sensors and EEG/sEMG acquisition system interfaces. In the UART1 port of the microcontroller, an ASK transmitter/receptor module enables other applications, such as the communication between two STIMGRASP stimulators, increasing the number of channels.

#### 2.2.5. The Microcontroller’s Firmware

The circuit of the STIMGRASP has the EEPROM coupled with the same I2C bus as that of the DAC converter, whose objective is to store the amplitude sequence time parameters. After the initialization, firmware executes an infinite loop, checking the battery status and for commands to activate the stimulator or to store new stimulation pulse parameters in the EEPROM. Thus, the STIMGRASP only needs to receive commands to activate the desired grasp modes once the stimulus parameter values are already stored. When the system starts, it reads all parameters and transfers them to the RAM of the microcontroller to generate the stimulation pulses. For smoother movements and user comfort, an amplitude modulation of the output channel signals is implemented with upward or downward ramps with configurable time lengths, adjusted independently for each channel.

### 2.3. Configuration and Control Platforms

Health professionals have a C# language software platform for the Windows operating system to set up STIMGRASP. This platform presents a single-screen interface, allowing an easy customization of various sequence times and amplitudes of the four preprogrammed stimulation sequences (hand opening, palmar and lateral grasping, and index finger extension) [37,38] and the programming of other stimulation sequences.

Moreover, neuroprosthesis users have a mobile app for Android with all stimulation commands. Since users have no hand dexterity, human–computer interaction usability concepts [39] were considered. Colors, confirmation messages, a minimum number of screens, and command buttons with enough sizing and spacing enable their activation and make the operation easy. Figure 7 shows some of the mobile app screens.

### 2.4. Orthosis

Considering the need for accurate electrode positioning, the orthosis completes the system (Figure 8). It uses a mesh structure allowing perfect anatomical adjustment after modeling and forearm ventilation. Conductive reusable rubber electrodes with foam blocks force them against the skin, and conductive gel makes the contact area uniform. Neodymium magnets fix the electrodes in the proper position, defined by the health professional during the device configuration process, leaving it ready for use.

### 2.5. System Validation

The system validation was carried out through a pilot experiment, in a single session, with a subject with a C6 SCI for nine years at the Movement Studies Laboratory/FMUSP, along with a health professional. An able-body subject also performed the tasks for reference. This trial complied with the tenets of the Declaration of Helsinki, and informed consent was obtained from the participants.

The following was evaluated:Donning and doffing the orthosis.The maintenance of the electrode positioning after configuration by the health professional and orthosis donning.Stimulus sequence configuration on the health professional’s platform.Mobile application command by the user.Generation and application of stimulus sequences for hand opening, palmar and lateral grasping, and forefinger extension.

The stimulation sequences were those established in [37,38] and shown in Table 1, Table 2 and Table 3. The selected muscles were: extensor carpi radialis (ECR), extensor digitorum communis (EDC), flexor digitorum superficialis (FDS), lumbricalis (L), abductor pollicis brevis (AbPB), and opponens pollicis (OpP). The health professional first defined electrode positioning, configured the amplitudes and sequences timing on the Windows platform, and finally, the user commands stimulation on the mobile app.

## 3. Results

### 3.1. Hardware Assembly

The complete final circuit of the STIMGRASP stimulation consisted of a two-sided board with dimensions of approximately 100 mm × 39 mm × 25 mm (length, width, height). With an optimized layout to reduce the size of the set, an additional board was positioned orthogonally to the mainboard, which contained the Bluetooth module and the status LEDs. As the battery used was relatively large, this board had a similar size, adding little to the volume of the set (Figure 9).

### 3.2. Battery Life

The battery life was verified in three situations of continuous use, changing the circuit consumption based on the output pulse amplitude:Moderate use: In this situation, the sequences of palmar grasp, lateral grasp, and forefinger extension (each of them with a 20-s duration) were alternated every 40 s, with a stimulation intensity between 4.8 and 6.6 mA. In that case, the battery lasted 14 h and 30 min until the automatic shutdown of the stimulator (when it reached 3.03 V).Intense use: In this situation, the sequences of moderate use were repeated with three times more stimulation intensity (from 15 to 20 mA). In that case, the battery lasted 13 h and 30 min until the automatic shutdown of the stimulator.Maximum use: In this case, all channels were considered with the maximum output amplitude (40 mA), resulting in a total current battery consumption of 440 mA. In that situation, the battery lasted 5 h and 40 min.

The case named intense use, considering 15–20 mA, is the commonly used amplitude. If continuous use were possible, the battery would last 13 h and 30 min until the automatic shutdown of the stimulator. However, under this condition, the muscle fatigues. It needs time intervals to prevent fatigue or to recover. Thus, the total time available to use the system is greater.

### 3.3. Output Pulses Generation

Figure 10A details a single biphasic output pulse with an amplitude of 38 mA and a pulse width of 300 µs/phase without considering the derivative compensation for skin characteristics, with a purely resistive load. Furthermore, Figure 10B shows the biphasic output pulse, with an amplitude of 34 mA and pulse width of 300 µs/phase, but with the compensation feedback under a charge of 1 kΩ, in parallel with a capacitor of 100 nF, modeling the human tissue [35]. Figure 10C presents a pulse train behavior of an output channel with an upward ramp of 3 s, followed by 2 s of stability and a downward ramp of 1 s. The number of amplitude steps to go upward or downward the ramp was calculated from the desired time interval and the amplitude variation specified in the health professional interface (in the case of Figure 10C, it ranged from 0 to 40 mA). Figure 10D details the 20 Hz pulse frequency, while Figure 10E depicts the eight multiplexed pulses. Each pulse corresponded to an independent amplitude setup with a time interval of approximately 2.5 ms between them, based on the times of relay switching and the total pulse width. This pulse train was repeated every 50 ms so that the stimulation frequency of each channel was 20 Hz.

### 3.4. System Validation

The SCI subject could don and doff the orthosis without assistance, as shown in Figure 11. However, the electrodes moved, and obtaining the desired movements with the stimulation was impossible. Thus, the evaluation of the stimulator was carried out using self-adhesive electrodes. The able-body subject could easily don and doff the orthosis without electrode movement. Figure 12 shows the resulting movements with the applied sequences for both subjects. The SCI subject could not perform lateral grasp.

## 4. Discussion

This study presented the STIMGRASP, a noninvasive neuromuscular functional electrical stimulator to be used as an assistive technology in ADLs. It allows for up to eight muscles to be stimulated in programmable sequences for functional hand movements, such as hand opening, palmar and lateral grasping, and forefinger extension. It is a battery-powered portable stimulator that is more secure and protects the user against hazardous current loops compared to other devices connected to the grid. Moreover, the STIMGRASP is a small unit that can be attached to the orthosis with the electrodes and is controlled wirelessly by a smartphone app, providing more mobility to the system and being more functional and feasible to perform daily activities. Therefore, the development system targets of portability, wearability, and user-friendliness were achieved beyond the fact that the STIMGRASP could also be used in other contexts, such as research applications and therapy sessions.

Except for the ReGrasp and Bionic Glove, those with the control unit attached to the orthosis, and the H200 presented as an orthosis with a small wireless control unit, other devices are presented as bigger separate units. However, the first ones provide only hand opening and closing. In this respect, Myndmove provides a complete stimulation protocol, allowing, in addition to the palmar and lateral grasps, pinch, tripod, and lumbrical grasps, and varied reaching movements with the eight channels.

The nominal current of the 3100 mAh rechargeable lithium-ion battery was superior to that of the other devices found in the literature (750 mAh for INTFES [18] and 350 mAh for Bioness H200 [40]), allowing the device to be used throughout the day (more than 10 h with continuous intense use) and its charging during the night. The prototype from Wang et al. [32] used a 3000 mAh battery, but no consumption tests were presented. It was observed that the increase in the stimulation current did not proportionally reduce the battery duration time since the voltage converter efficiency improved. Additionally, since the width of the stimulation pulses was reduced (300 µs) concerning the interval between pulses (50 ms), most of the current consumption was constant, regardless of the amplitude of the current pulses generated.

As the battery presented a low voltage (3.7 V for the VNCR18650A), it required a buck–boost converter to power the different circuits and generate current amplitudes to stimulate the human tissue. The project used simple and compact buck–boost controllers designed for DC–DC conversion topologies, where low cost, small size, and voltage requirements were top priorities. These devices operate on frequencies of hundreds of kHz to reduce the size of magnetic elements, achieving good efficiency (at least 85%). Commercial solutions were also implemented by Xu et al. [29] and Malešević et al. [18].

A symmetric biphasic constant current waveform with pulse characteristics in the commonly used range was adopted. The frequency and pulse width were fixed at 20 Hz and 300 µs/phase, respectively, while the amplitude could be adjusted up to a maximum of 40 mA (load of 1 kΩ) as well as providing modulation timing. While minimizing muscle fatigue, increasing comfort, and ensuring muscle contraction with smoother movement transitions, it facilitated and sped up device configuration.

The most commonly used circuit to generate the symmetrical biphasic waveforms was the H-bridge switching circuit, which activates a pair of transistors at a time so that the current goes through the load in both directions. The literature has shown its implementation combined with Wilson current mirror circuits, voltage-controlled current sources, and voltage-to-current converter [18,27,28,30,32]. Despite being efficient for a single channel, these circuits use many components, making the replication for each channel with equal response challenging work. Masdar et al. [31] gave a different solution, using a voltage-to-current converter based on a power operational amplifier to generate the output pulses. Instead, the circuit presented here used a modified Howland current source topology. It presented three advantages compared with the previous solutions: it was a simple implementation with a smaller number of components, allowed a fast switching for channel sequencing, and had a small standby current consumption (few milliwatts compared to a maximum of 1.08 W of LM675). Compared to the traditional H-bridge circuit, a disadvantage of the presented topology is the need for a symmetrical source.

The method to activate several channels in sequential mode demanded reduced response times, a stabilization of the current source, and resulted in the miniaturization of the circuit, similar to those proposed by Wang et al. [32] and Malešević et al. [18]. This strategy needed a single output circuit, while other solutions must replicate it by the number of channels. However, different components were implemented. Wang et al. [32] used the MAX14803 available in a 48-pin LQFP package, 16-channel high-voltage analog switches (single pole, single throw) with internal digital control. Instead, the circuit implemented here used eight CPC2030N with two independent, normally open, solid-state relays in an eight-pin SOIC package that employs optically coupled MOSFET technology. The last solution took up more space on the PCB board but guaranteed optical isolation.

As expected, the output signal for a purely resistive load without the compensation circuit detailed in Figure 10A matched the digital pulse configuration. However, considering an RC load, which better represented the electrode–tissue impedance [35], the output signal presented delays when increasing or decreasing, such as a charge and discharge of a capacitor. Thus, a derivative compensation negative feedback (C9/R11 and C53/R49) accelerated the circuit response, partially correcting the system response for a critically dampened behavior. Figure 10B showed the proposed circuit response, which approximated that of Figure 10A. However, there was still a small charge imbalance due to pulse width differences between pulse phases, requiring further adjustments. The compensation is a complex solution since the load value is unknown.

The USB–serial converter/Bluetooth modules will allow the STIMGRASP to be controlled in several ways in future applications, whether directly from a computer or smartphone, as presented here, or even from an acquisition interface such as EEG (BCI), EMG, among others, as briefly presented in [7]. Bluetooth communication provides isolation from the grid and remote control of the device using any device host with a standard Bluetooth module. Malešević et al. [18] also proposed this type of communication, while the Bioness H200 used radiofrequency to communicate with the control unit.

The orthosis with the electrodes is a crucial part of the system to achieve the wearability prerequisite. Although self-adhesive electrodes are the most commonly used, conductive reusable rubber electrodes are an attractive option since they can be used much longer than conventional adhesives. Nonetheless, there is a need for uniform skin contact provided by conductive gel and fixation system.

Although the SCI subject could easily don and doff the orthosis, he needed to press it against the wall for closure (Figure 11). A more flexible structure using 3D printing could be an alternative to this proposal. The foam ensured a mechanical electrode–skin contact, but the electrodes moved, requiring stronger magnets to fix them, preventing displacement during orthosis donning. In the case of a 3D-printed orthosis, electrode positioning could be customized during the design processes. On the other hand, the able-body subject could wear the orthosis without moving the electrodes.

Thus, adhesive electrodes were used for the SCI subject, enabling the application of stimulation sequences and the evaluation of the interfaces, while for the able-body subject the orthosis was used. Figure 12 showed the resulted movements for the applied stimulation sequences for both subjects, showing the adequacy of the generated signals. Although the SCI subject presented some joint stiffness due to the absence of physical therapy for several years and the fact that it was a pilot test in a single session, functional movements of hand opening, palmar grasping, and forefinger extension could be obtained. The lateral grasping was not immediately achieved as the other patterns.

The health professional and SCI subject could configure and control the STIMGRASP through the software platforms, which generated the hand opening, palmar and lateral grasping, and forefinger extension sequences. Although only a pilot experiment was conducted for system validation, it is reasonable to consider that people with tetraplegia and hemiplegia already use smartphones for daily tasks, even if in an adapted or limited way. Furthermore, a smartphone is a versatile tool with an excellent processing capacity, enabling interfaces with resources adapted to the user’s limitations. Thus, it is believed that the smartphone provides advantages over traditional methods, enabling the creation of user-friendly interfaces. The smartphone app used large and spaced buttons, apart from text boxes with additional information that was easy to interpret.

Several applications used buttons as interface control, but none used smartphone apps. The NESS Handmaster and its posterior evolution, the Bioness H200, are examples of this kind of system [13,40]. The presented hardware solution foresees the application of other control interfaces from the USB–serial converter/Bluetooth communication modules, although the software platform in its current version does not consider them yet. The Compex Motion and the Motionstim 8 also allow various man–machine interfaces, such as the one used by Kapadia et al. [9]. However, in the applications proposed by Mangold et al. [15] and Popovic and Kekker [14] with the Compex Motion, they used a button control, while Rupp et al. [16] used the Motionstim 8 with a shoulder position sensor. The Bionic Glove [12] also used a position sensor to detect voluntary wrist movement and trigger the stimulation sequences.

Moreover, there is evidence of a small amount of preservation across the injury, even in clinically complete SCI subjects, which is insufficient to produce functional movements [41,42,43]. Furthermore, there is also evidence that FES application can enhance neuroplasticity in SCI subjects and those with hemiplegia, helping them regain some voluntary functions [9,44]. Therefore, using user interfaces based on EMG or EEG signals to guarantee the synchronization between stimulation delivery and active user participation seems to be an important issue, allowing subjects to attempt each movement while FES is applied to assist movement completion. Such associative interventions, combining cortical activation and peripheral stimulation, induce the nervous system’s experience-dependent reorganization.

Kapadia et al. [9] showed the benefits of the clinical application of FES to train upper-extremity function for over 15 years, helping over 200 stroke or SCI patients, stating the need for subject participation in movement execution. Argentim et al. [19] proposed a system that used a threshold sEMG to trigger the stimulation for people with hemiplegia. The threshold was defined based on the subject’s capability and guaranteed the movement initiation by the subject and movement completion by the FES system. BCI technologies offer a direct way to synchronize cortical commands and movements generated by FES, which can be advantageous for inducing neuroplasticity and has gained much attention. However, despite last year’s efforts, BCI systems have not yet reached everyday applications due to their complexity [7,21,22,23,24,25].

## 5. Conclusions

The STIMGRASP system, consisting of a noninvasive portable energy-efficient electrical stimulator, an orthosis, and control software, is focused on daily use as an assistive technology. The relevance of the system is highlighted by the following aspects:1.SocialThe system aims to reinsert individuals with SCI and hemiplegia in society by increasing their independence to carry out ADLs without aid from other people.2.TherapeuticElectrical stimulation keeps the muscles conditioned, avoiding muscular atrophy and joint stiffness.Although the system is focused on executing functional grasping as an assistive device with a mobile app control platform, it can also be used in therapy sessions with the same software program and a Windows operating system platform for configuration and therapy.3.FunctionalThe system successfully provided hand opening, palmar and lateral grasping, and forefinger extension movements. However, since it was validated in a single-section pilot experiment, clinical trials must be done to complete system evaluation.The USB–serial converter/Bluetooth modules communication allowed other sensors to be connected to the system, increasing the number of control possibilities.The orthosis allowed system and electrode fixation, all in one piece, qualifying as an assistive device for ADLs. Although the orthosis was easy to don and doff, the electrodes moved during orthosis donning, requiring improvements.The battery life allowed the system to be used during the day and charged at night.4.TechnicalThe system provided eight stimulation channels, with more efficient circuits and modern components, in a smaller physical size.A derivative compensation negative feedback was proposed to accelerate the circuit response considering an RC load, which better represented the electrode–tissue impedance, partially correcting the system response for a critically dampened behavior. However, results showed a small charge imbalance, requiring further adjustments. The compensation is a complex solution since the load value is unknown.The developed system was characterized as an open-loop system (there was no feedback for the user regarding the force applied in the generated grasp). However, there are additional circuits expected in the hardware platform, which could transform it into a closed-loop system.

## Figures and Tables

**Figure 1 sensors-23-00010-f001:**
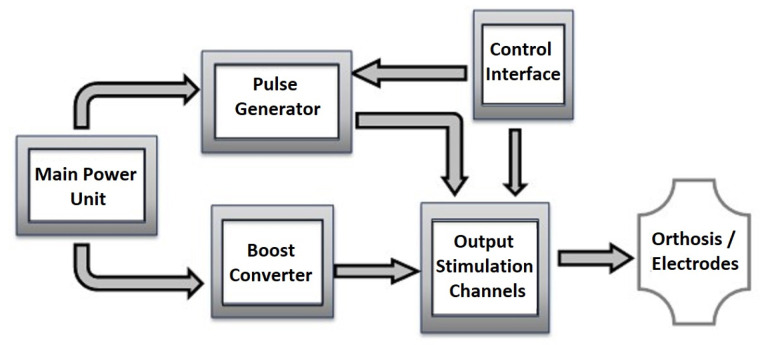
Generic block diagram of a stimulator.

**Figure 2 sensors-23-00010-f002:**
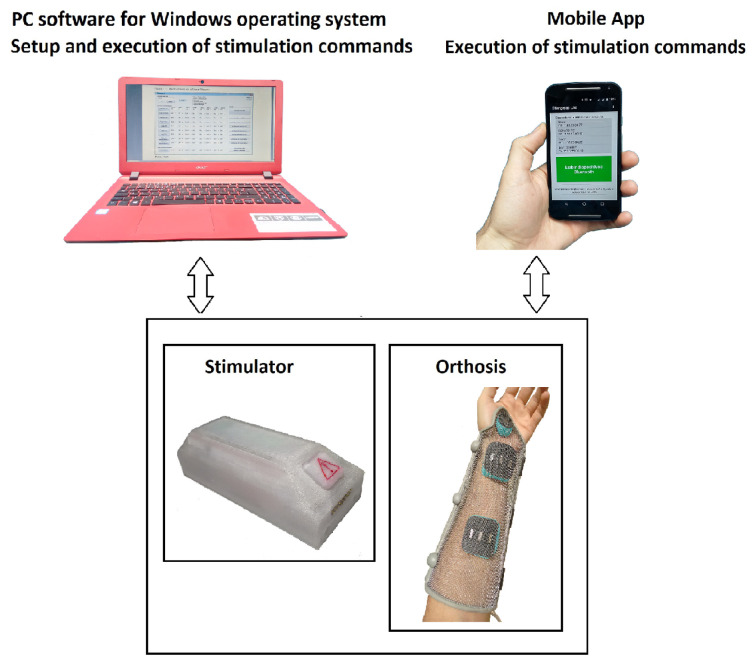
Main elements of the system.

**Figure 3 sensors-23-00010-f003:**
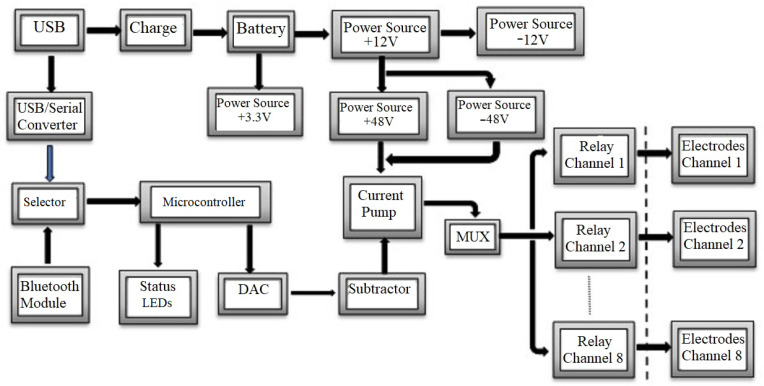
Block diagram of the STIMGRASP stimulator.

**Figure 4 sensors-23-00010-f004:**
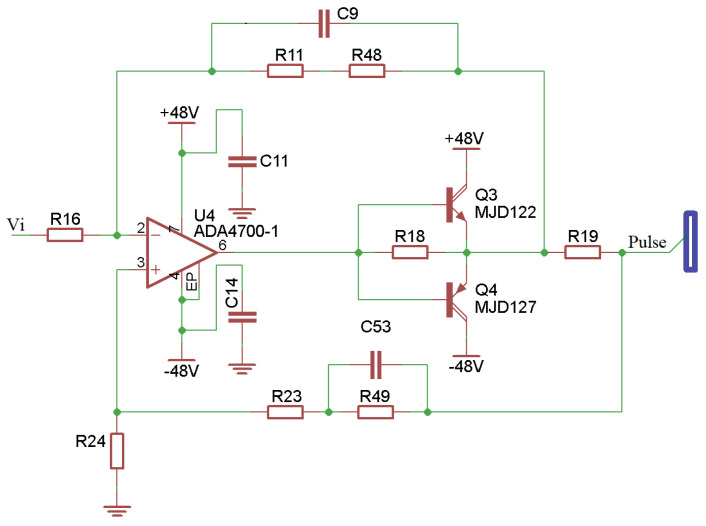
Modified Howland current source circuit.

**Figure 5 sensors-23-00010-f005:**
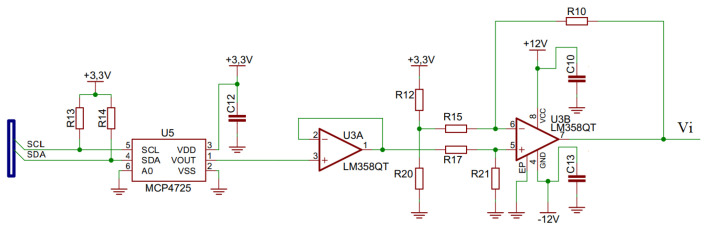
D/A converter and zero-reference adjustment circuit.

**Figure 6 sensors-23-00010-f006:**
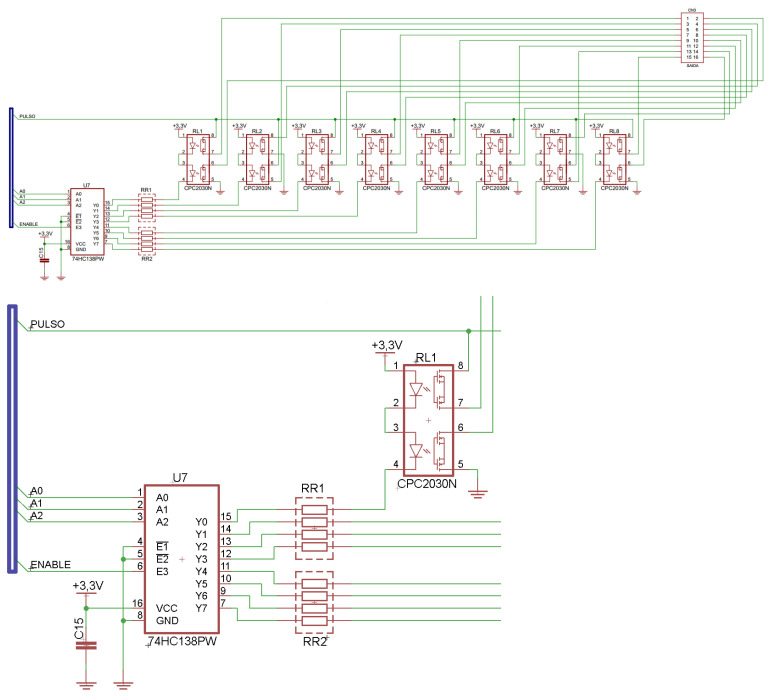
Multiplexing circuit with solid-state relays.

**Figure 7 sensors-23-00010-f007:**
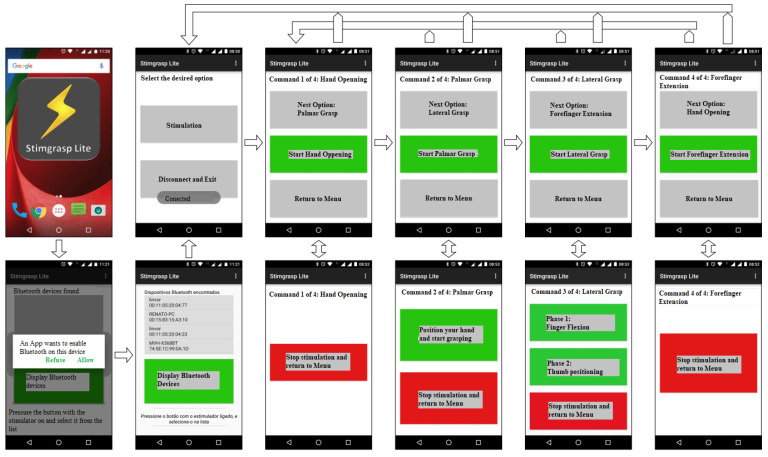
Mobile app screens.

**Figure 8 sensors-23-00010-f008:**
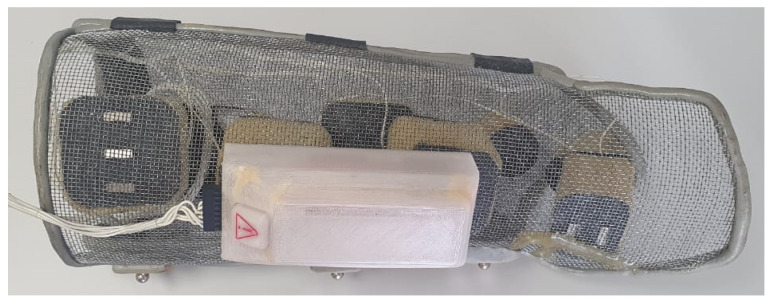
Orthosis with the electrodes and the STIMGRASP stimulator.

**Figure 9 sensors-23-00010-f009:**
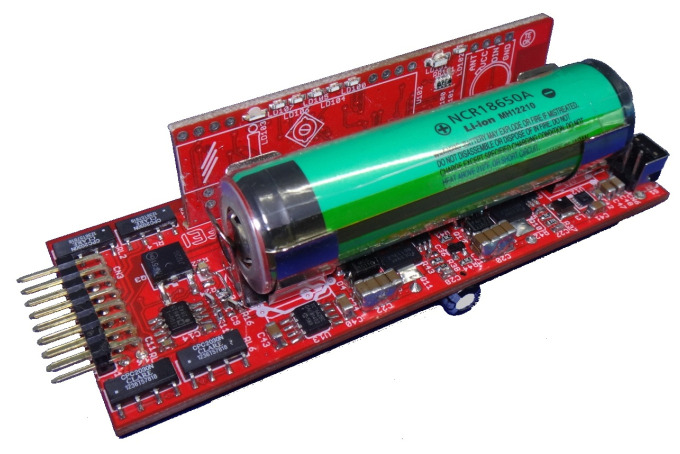
Hardware assembly.

**Figure 10 sensors-23-00010-f010:**
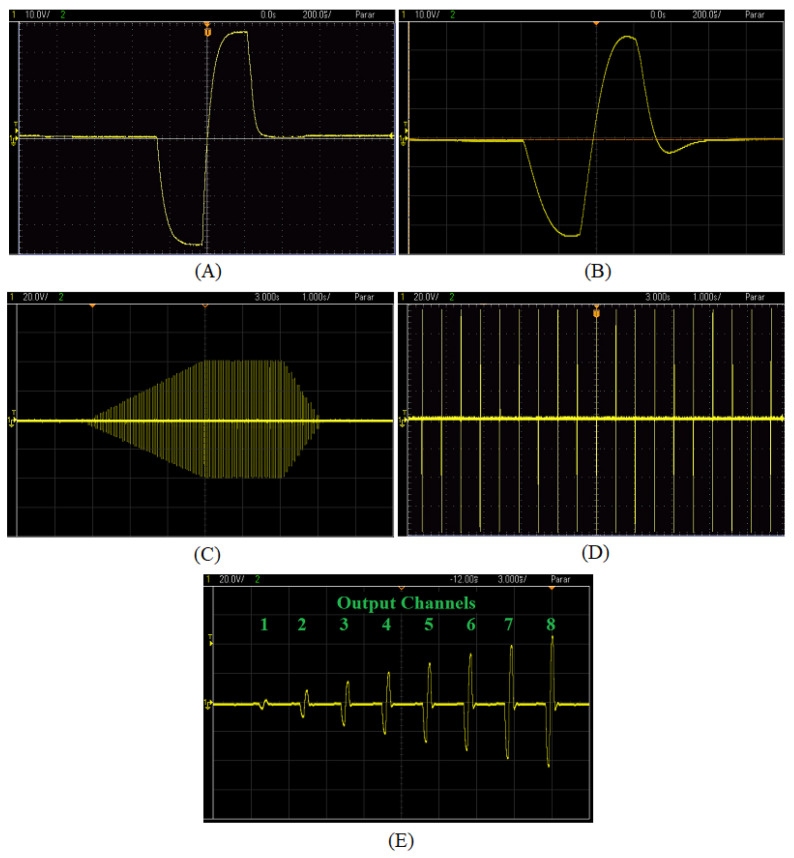
STIMGRASP output signal. (**A**) Output biphasic pulse with resistive load; (**B**) output biphasic pulse with RC load; (**C**) pulse train in a channel with upward and downward ramps; (**D**) pulse train in a channel; (**E**) eight multiplexed output channels.

**Figure 11 sensors-23-00010-f011:**
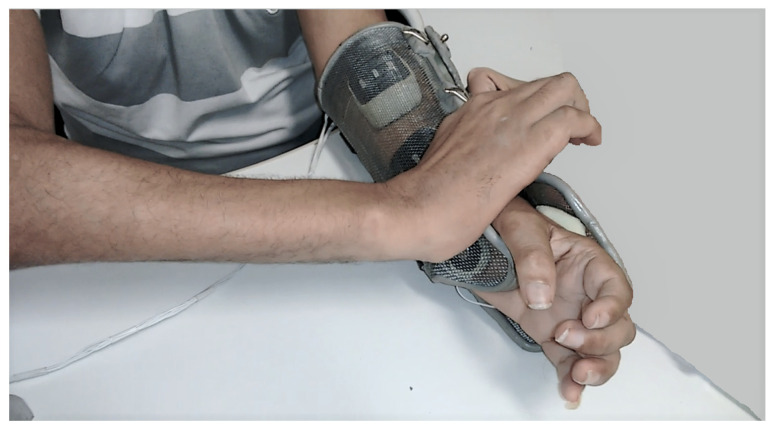
SCI individual donning the orthosis.

**Figure 12 sensors-23-00010-f012:**
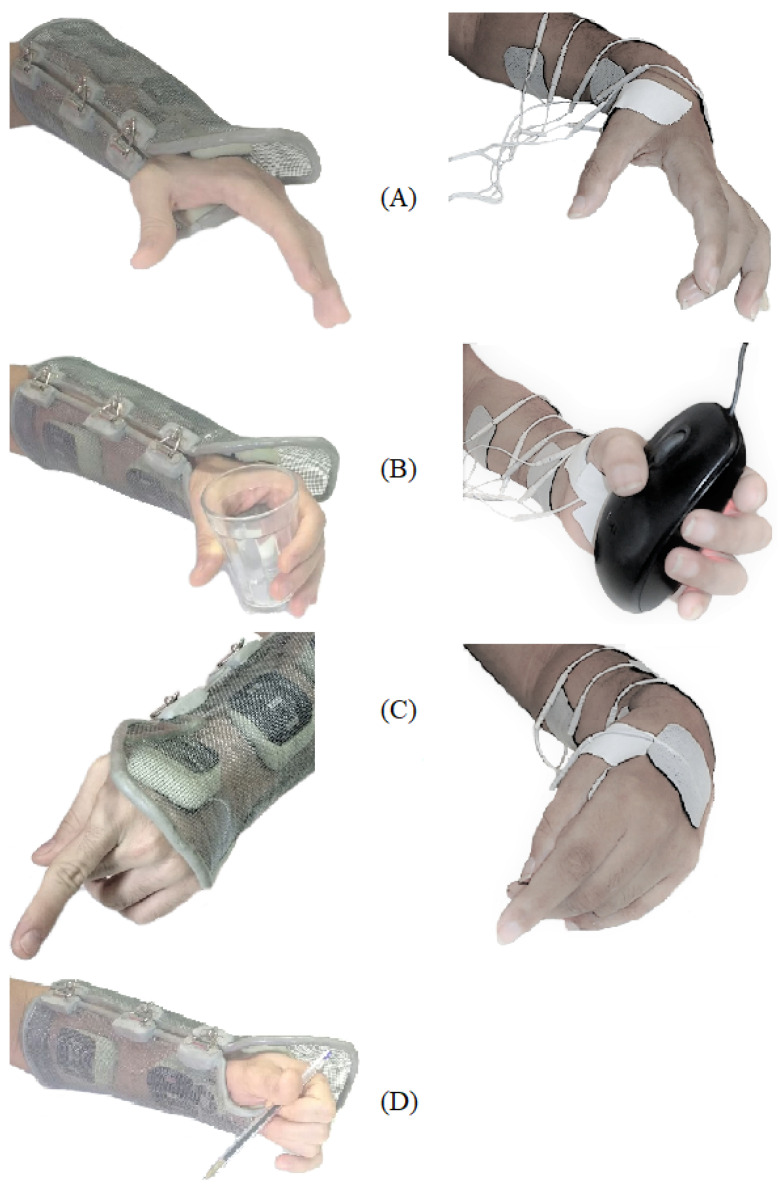
Movements with the applied sequences—able-body individual on the left and SCI individual on the right. (**A**) Hand Opening; (**B**) palmar grasping; (**C**) forefinger extension; (**D**) lateral grasping.

**Table 1 sensors-23-00010-t001:** Stimulation sequence for palmar grasping.

Subphases	Muscles				
Opening	ECR	EDC	AbPB		
Positioning	ECR	AbPB	L		
Grasp	ECR	AbPB	L	FDS	OpP
Releasing	ECR	EDC	AbPB		

**Table 2 sensors-23-00010-t002:** Stimulation sequence for lateral grasping.

Subphases	Muscles			
Opening	ECR	EDC	AbPB	
Positioning	ECR	AbPB	FDS	
Grasp	ECR	AbPB	FDS	OpP
Releasing	ECR	EDC	AbPB	

**Table 3 sensors-23-00010-t003:** Stimulation sequence for forefinger extension.

Subphases	Muscles		
Opening	ECR	EDC	AbPB
Positioning	L	FDS	
Opening	ECR	EDC	AbPB

## Data Availability

Not applicable.

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
