# Peer review of "STIMGRASP: A Home-Based Functional Electrical Stimulator for Grasp Restoration in Daily Activities"

_sensors, 2022, doi:10.3390/s23010010_

Round 1

Reviewer 1 Report

The manuscript presents the design and implementation of a home-based functional electrical stimulator with eight output channels in the open-loop configuration. The motivation and the current state are comprehensively described in the manuscript. The structure of the manuscript is well chosen, and clearly stated outcomes in Discussion section are supported by results. I would really appreciate to include full color images as in [35]. After minor corrections of typos (Fig 3 – Selector, Microcontroller, LEDs; missing Ref. no 23.), I recommend accepting the manuscript.

Author Response

Dear Reviewer,

The authors are grateful for the reviewer's appreciation and feedback. We have revised the manuscript and addressed the reviewer's comments. The revised manuscript highlights the changes which have improved the paper. 

  1. We rephrased the text and corrected the spelling, improving English writing.
  2. The Introduction section was reorganized. The subsection Basic Stimulator Architecture was created to separate the general background of FES, older and commercial FES devices, to the main electronic characteristic implementations.
  3. Full-color images were added as required.
  4. As requested, we added System Validation's subsection in the Material and Methods and Results sections. It defines a pilot experiment to verify the practical system's functioning, highlighting the proposed movements performed by a spinal cord injury individual in contrast with an able-body individual.
  5. The discussion section was also improved, comparing the proposed system features with those presented in the literature, and the  Conclusion section highlighted the work's essential points.

Reviewer 2 Report

The work presented in this paper seems interesting and makes a good introduction to the problem, but I have several concerns that should be clarified:

1. As the authors themselves state in discussion, the study presents the electronic circuit of STIMGRASP, but I find necessary to carry out at least some pilot test on patients that do not seem to have been carried out. I find it should be performed and results added. If not at least it should be stated in tittle and clearly claimed that limitation of the work in discussion.

2. In the introduction calls a FES system used as assistive technology as a neurprosthesis, and when defining it, it is said that in some cases it includes an orthosis that provides additional assistance to perform movements, and it could be inferred that are assistive orthosis, not  only to place the electrodes for stimulate the muscles. It should be clarified.

3. It is referred the BionessH200 as the only neuroprosthesis currently commercialized. The paper says as novelties of the system proposed the low-cost, the conjunction of several technical improvements related in literature and the functionality since functional grasps are said to be performed. It seems that cylindrical and pinch grasps are said to be feasible. As for low-cost it would be interesting to know the economic improvement with regard to the commercialized model and if it is going to be commercialized, but again, first, it should be tested in patients. And with regard to grasps, it talks of four muscular activation: hand opening, palmar and lateral grasp and forefinger extension. It should be clarified the stimulating signals that should be provided to perform these movements. Furthermore, it should be clarified if, in case of success in the performance of the movement, these would allow to perform stable grasps.

3. He talks about the importance of electrode placement, which is obvious, but is left to the 'experience' of the clinician. I find it should be clarified how many electrodes are and which muscles they pretend to activate for each movement. Not all subjects perform grasps with same activation, does the device need a custom configuration?

4. At the end of discussion it is said  'However, the use of EMG to ensure synchronization between stimulation and active user participation appears to be an important issue, especially for hemiplegics' Please clarify how EMG is going to be used in the device.

Author Response

Dear Reviewer, 

The authors are grateful for the reviewer's appreciation and feedback. We have revised the manuscript and addressed the reviewer's comments. The revised manuscript highlights the changes which have improved the paper. 

  1. We rephrased the text and corrected the spelling, improving English writing.
  2. The Introduction section was reorganized. The subsection Basic Stimulator Architecture was created to separate the general background of FES, older and commercial FES devices, to the main electronic characteristic implementations.
  3. Full-color images were added as required.
  4. As requested, we added System Validation's subsections in the Material and Methods (subsection 2.5 on page 12) and Results (subsection 3.4 on page 14) sections. The first defines a pilot experiment to verify the practical system's functioning, including the stimulation sequences. At the same time, the latter presents the validation results, highlighting the proposed movements performed by a spinal cord injury individual in contrast with an able-body individual.
  5. The discussion section was also improved, comparing the proposed system features with those presented in the literature, and the  Conclusion section highlighted the work's essential points.
  6. We agree with the reviewer that a complete evaluation is needed in a clinical trial. The presented work describes the system development, especially the hardware, and its application will be the focus of another work.
  7. It was clarified in lines 48-50 that the orthosis provides additional mechanical assistance and helps to fix the electrodes. In lines 60-69, more details about the Bionic Glove and its subsequent models, including a currently commercialized version - ReGrasp, were added. In the next paragraph, the same was provided about Ness Handmaster and the current version, H200. These systems are examples of neuroprosthesis.
  8. As mentioned before, the requested stimulation sequences were provided in subsection 2.5 - System Evaluation. These sequences were previously defined and applied in a small clinical trial showing movement functionality (references 37 and 38). Although the stimulator has eight channels, in these sequences were stimulated up to 5 muscles, depending on the desired movement. However, the use of the orthosis may suppress the need for the extensor carpi stimulation.
  9. Regarding the cost comparison, we could not obtain the market value of the H200. However, the ReGrasp is commercialized at $2995, as shown on the dealer's website. STIMGRASP is still in the academic development phase, and few units have been made. The cost per unit is between US$200 - US$300, but we need the future commercial sale value. For this reason, we excluded the cost approach from the text.
  10. The system needs a custom configuration. Based on our experience, although there are key points for positioning the electrodes, the best performance of the resulting movements depends on fine-tuning not only the positioning of the electrodes but also the stimulus parameters; in the case of the proposed system of amplitude and timing.
  11. The two known commercial neuroprostheses, the ReGrasp and the H200, use buttons to trigger the stimulation sequences. The first still provides an earing sensor that controls stimulation from head movement. However, using user interfaces based on EEG or EMG to ensure synchronization between stimulation and active user participation allows subjects to attempt each movement while FES assists in movement completion. Such associative interventions, combining cortical activation and peripheral stimulation, induce the nervous system experience-dependent re-organization. The Discussion section (in lines 588-591 and 633-635) now includes the information that the presented hardware solution foresees the application of other control interfaces from the USB-Serial converter/ Bluetooth communication modules, although the software in its current version does not consider them yet. However, a simple application can be the implementation of a customized sEMG threshold to trigger the stimulation sequence.

Round 2

Reviewer 2 Report

All my concerns have been adressed. Only the definition of muscle abbreviations is missing in captions of tables 1 to 3.

Author Response

Dear Reviewer, 

The authors are grateful for the reviewer's appreciation and feedback. We apologize for forgetting to include abbreviation definitions. We have revised the manuscript and addressed the reviewer's comment.

The definitions of muscle abbreviations have been added. However, to avoid repetition, the definitions were added in the text instead of added in the tables chapters.

Paragraph on line 311 is now written as:
The stimulation sequences were those established in [37,38] and shown in Tables 1 to 3. The selected muscles were: Extensor Carpi Radialis (ECR), Extensor Digitorum Communis (EDC), Flexor Digitorum Superficialis (FDS), Lumbricalis (L), Abductor Pollicis Brevis (AbPB), and Opponens Pollicis (OpP). The health professional first defines electrode positioning, configures amplitudes and sequences timing on the Windows Platform, and finally, the user commands stimulation on the mobile App.